# Efficient $n$-body simulations using physics informed graph neural networks

**Víctor Ramos Osuna**
Departmento de Sistemas Informáticos
E.T.S.I Sistemas Informáticos – Universidad Politécnica de Madrid
C. de Alan Turing, s/n, 28031 Madrid, Spain
victor.ramos.osuna@upm.es

**Alberto Díaz-Álvarez**
Departmento de Sistemas Informáticos
E.T.S.I Sistemas Informáticos – Universidad Politécnica de Madrid
C. de Alan Turing, s/n, 28031 Madrid, Spain
alberto.diaz@upm.es

**Raúl Lara Cabrera**
Departmento de Sistemas Informáticos
E.T.S.I Sistemas Informáticos – Universidad Politécnica de Madrid
C. de Alan Turing, s/n, 28031 Madrid, Spain
raul.lara@upm.es

## Abstract

This paper presents a novel approach for accelerating $N$-bodies simulations by integrating a physics-informed graph neural networks (GNNs) with traditional numerical methods. Our method implements a leapfrog-based simulation engine to generate datasets from diverse astrophysical scenarios which are then transformed into graph representations. A custom-designed GNN is trained to predict particle accelerations with high precision. Experiments, conducted on 60 training and 6 testing simulations spanning from 3 to 500 bodies over 1000 time steps, demonstrate that the proposed model achieves extremely low prediction errors—loss values while maintaining robust long-term stability, with accumulated errors in position, velocity, and acceleration remaining insignificant. Furthermore, our method yields a modest speedup of approximately 17% over conventional simulation techniques. These results indicate that the integration of deep learning with traditional physical simulation methods offers a promising pathway to significantly enhance computational efficiency without compromising accuracy.

## 1   Introduction

A $N$-bodies system models the evolution of $N$ particles under mutual gravitational interactions, governed by Newton's laws. Analytical solutions exist only for $N = 2$; for larger $N$, numerical methods are required. Despite their precision, these approaches are computationally expensive due to the need to evaluate every pairwise interaction at each step Aarseth and Aarseth (2003), motivating the search for faster alternatives that preserve accuracy.

Over the past two decades, deep learning (DL) techniques have opened new avenues for modelling such complex systems. Among them, graph neural networks (GNNs) stand out for their ability

XVI XVI Congreso Español de Metaheurísticas, Algoritmos Evolutivos y Bioinspirados (maeb 2025).

to capture both local and global relationships in graph-structured data, as demonstrated in various physical prediction tasks Battaglia et al. (2018). Although gravity is a long-range force, we adopt a local interaction approximation—similar to hierarchical methods like Barnes-Hut Barnes and Hut (1986)—to improve efficiency. While this simplifies the problem, it enables the GNN to approximate global dynamics effectively.

In this work, we propose an approach that combines the generation of simulation data from an $N$-bodies system with a regression model based on GNNs. The methodology presented consists of three main modules:

1. **Physical simulation and data generation**: A simulation engine employing a leapfrog integration scheme Iserles (1986) iteratively updates the positions, velocities, and accelerations of particles. It generates diverse scenarios (e.g. random distributions, 3D discs, multi-disc systems) to create a dataset containing both particle attributes and their target accelerations.
2. **Transformation to graph representation**: The simulation data is converted into graphs, with each particle represented as a node (including attributes such as position) and edges established based on proximity.
3. **GNNs-based regression model**: A graph neural network based on the *EdgeConv* architecture Wang et al. (2019) propagates and transforms node information to predict particle accelerations. The model includes encoders for nodes and edges, mapping initial features into a higher-dimensional space, and an output layer tailored for the regression task.

The **primary objective** of this work is to **validate the ability of the GNN model to predict accelerations in $N$-bodies systems with sufficient precision**, demonstrating that the integration of realistic physical simulation with DL can substantially improve the simulation time of complex dynamics while maintaining a high degree of fidelity to reality.

The remainder of the paper is organised as follows. Section Related works reviews recent work on the application of DL in computational physics. This is followed by Sections Methodology and Experiments and results, which detail the structure, execution, and results of the experiments. Finally, the paper concludes with Section Conclusion, which presents the findings and outlines potential future directions for this work.

## 2   Related works

The simulation of $N$-bodies systems has long been a core topic in computational physics, especially in astrophysics. Early work from the 1980s relied on direct methods to compute gravitational interactions between all particle pairs. While precise for small systems, these algorithms have $O(N^2)$ complexity, which scales poorly. This led to the development of more efficient alternatives, notably the Barnes-Hut method Barnes and Hut (1986), which reduces complexity to $O(N \log N)$ using a hierarchical approximation of forces, and the fast multipole method (FMM) Greengard and Rokhlin (1987), which achieves $O(N)$ by grouping distant particles into clusters.

In parallel, numerical integration schemes were developed to simulate system evolution over time. The leapfrog integrator, introduced in 1967, remains one of the most widely used methods due to its ability to preserve physical invariants such as energy Iserles (1986). These techniques have enabled simulations of stellar dynamics and galaxy formation Hernquist (1990); Navarro et al. (1997), often using parametric density profiles to approximate mass distribution.

Since the 2010s, the rise of machine learning (ML) has encouraged the combination of traditional methods with artificial intelligence (AI) techniques. Sánchez-González et al.Sanchez-Gonzalez et al. (2020) showed that GNNs can learn latent representations that capture system dynamics across scales, enabling accurate predictions of trajectories and accelerations without explicitly computing all interactions. This significantly reduces computational cost, allowing application to scenarios where full simulation would be prohibitive, such as diffusion, turbulence, or granular systemsRaissi et al. (2019); Pfaff et al. (2020). Similar successes have been reported in related areas like fluid dynamics and materials modelling Maurizi et al. (2022); Sergeev et al. (2024).

GNNs have also proven highly effective for regression tasks in physics. They have been used to predict continuous quantities—such as forces, energies, or accelerations—with high accuracy Battaglia et al. (2016); Kipf and Welling (2016), thanks to their ability to capture both local structure and global

connectivity. This allows them to infer variations in force fields from particle distributions Gilmer et al. (2017), and generalise across different configurations, including materials and biological systems Chen et al. (2023); Yang et al. (2022); Bongini et al. (2022).

In summary, while classical models remain fundamental for $N$-bodies simulation, the literature shows that DL—and GNNs in particular—can ease their computational burden. Rather than replacing numerical methods (e.g., finite difference or finite element), GNNs serve as efficient computational shortcuts that complement physical models, providing data-driven approximations that reduce cost without compromising physical fidelity.

# 3    Methodology

We will divide this section into the four main stages that have comprised this study: (i) generation of a dataset from a physical $N$-bodies simulation, (ii) transformation of the raw dataset into a graph structure, (iii) definition of a GNN architecture for a regression problem, that is, the prediction of each body's acceleration, and (iv)training and evaluation of the model.

All experiments were executed on a workstation equipped with an AMD Ryzen 5 5600X 6-Core Processor (6 cores, 12 threads) running Ubuntu GNU/Linux 24.04, with a frequency of approximately 4650 MHz, and 32 GB of system memory. The machine uses an NVIDIA GeForce RTX 2070 graphics card with 8 GB of dedicated memory, running under CUDA 12.4 with driver version 550.120.

## 3.1    Data generation from physical simulation

The foundation of the methodology is a simulation engine that integrates the evolution of $N$-bodies systems. It begins with a set of initial conditions defined in terms of the particles' positions, velocities, and masses. The simulation engine employs a leapfrog integration scheme, which guarantees stable numerical integration and preserves system invariants (e.g. total energy) over time.

### 3.1.1    Gravitational acceleration formula

For each particle $i$ in a system of $N$ particles, the acceleration is computed by summing the contributions from all particles $j$ (excluding $i$, as is evident) as follows:

$$\vec{a_i} = G \sum_{j=1, j \neq i}^{N} m_j \frac{\vec{r_j} - \vec{r_i}}{\left( \|\vec{r_j} - \vec{r_i}\|^2 + \epsilon^2 \right)^{3/2}} \tag{1}$$

where $G$ is the gravitational constant, $m_j$ is the mass of particle $j$, $\vec{r_i}$ and $\vec{r_j}$ are the positions of particles $i$ and $j$ respectively, and $\epsilon$ is the smoothing parameter used to avoid singularities. The pseudocode is presented in Algorithm 1.

---

**Algorithm 1** $N$-Bodies simulation using leapfrog method

---

 1: **procedure** NBODYSIMULATION($R, V, M, dt, T, G, \varepsilon$)
 2:     $A \leftarrow$ Accelerations($R, M, G, \varepsilon$)
 3:     **for** $t \leftarrow 1$ **to** $T$ **do**
 4:         **for** $i \leftarrow 1$ **to** $N$ **do**
 5:             $V[i] \leftarrow V[i] + A[i] \cdot \frac{dt}{2}$
 6:             $R[i] \leftarrow R[i] + V[i] \cdot dt$
 7:         **end for**
 8:         $A \leftarrow$ Accelerations($R, M, G, \varepsilon$)
 9:         **for** $i \leftarrow 1$ **to** $N$ **do**
10:             $V[i] \leftarrow V[i] + A[i] \cdot \frac{dt}{2}$
11:         **end for**
12:         Store $(R, V, A)$
13:     **end for**
14: **end procedure**

---

In this algorithm, the function `Accelerations` is responsible for implementing the formula from Equation 1 for each particle.

### 3.1.2 Dataset generation

The dataset is generated from multiple simulations by varying the initial parameters (i.e. random distribution, $3D$ disc, multi-disc), organising the information into three sets:

- **Positions** $\{\vec{r_t}\}$ for each frame $t$.
- **Features**: These may include additional information (e.g. mass).
- **Regression labels** corresponding to the accelerations $\{\vec{a_t}\}$.

Additionally, the approach includes the possibility of incorporating temporal information by shifting the position sequence, which is achieved by concatenating previous frames as part of the feature vector for each time step.

## 3.2 Transformation to graph structure

The dataset is structured in a tabular format, but to capture the inherent relational structure of these $N$-bodies systems, we will transform it into a graph-based structure. To achieve this:

1. Each particle is represented as a **node**, with its initial feature being a vector that includes its position, its mass, and other data derived from its temporal evolution.

2. These nodes will be connected by **edges** based on a $k$-nearest neighbors (KNN) criterion, where each node is connected to its $k$ closest neighbors in terms of Euclidean distance. The parameter $k$ has been experimentally defined. Optionally, edge attributes may be computed, for example, the magnitude of the position difference: $e_{ij} = \|\vec{r_i} - \vec{r_j}\|$. However, in this work, no attributes have been incorporated into the edges.

Algorithm 2 describes the process by which each frame of the dataset is transformed into a graph.

---

**Algorithm 2** Frame to graph transformation using KNN

---

1: **procedure** GRAPHTRANSFORMATION($R, X, Y, k$)
2:  **for** $i \leftarrow 1$ **to** $N$ **do**
3:   Add node $i$ with feature $X[i]$
4:   Find $k$ nearest neighbours of $i$
5:   **for** each neighbour $j$ **do**
6:    Add edge $(i, j)$ with attribute $\|R[i] - R[j]\|$
7:   **end for**
8:  **end for**
9:  Build graph $G$ with assigned $Y$ tags
10:  **return** $G$
11: **end procedure**

---

## 3.3 Model architecture and inference

The model architecture is structured into three main modules: (i) the initial transformation of input features, (ii) the iterative propagation of information through message passing layers, and (iii) the generation of the final output for the regression task.

It has been designed to be modular, allowing experiments with different types of aggregation functions –such as mean or maximum–modifying the depth of the network, or incorporating attention mechanisms to weight the messages from neighbouring nodes differently.

All the implementation is based on PyTorch and PyTorch Geometric, leveraging vectorised operations and GPU acceleration, which is essential when training with large datasets and/or models.

Algorithm 3 describes the inference process through the proposed model.

**Algorithm 3** Inference process

---

1: **procedure** FORWARDPASS($G, L$)
2:     **for** node $i$ in $G$ **do**
3:         $h[i] \leftarrow \text{NodeEncoder}(X[i])$
4:     **end for**
5:     **for** $l \leftarrow 1$ **to** $L$ **do**
6:         **for** node $i$ in $G$ **do**
7:             $m[i] \leftarrow \sum_{j \in \mathcal{N}(i)} \phi\big(h[i], h[j], E[i, j]\big)$
8:             $h[i] \leftarrow \text{LayerNorm}\,(\text{concat}(h[i], m[i]))$
9:         **end for**
10:     **end for**
11:     **for** node $i$ in $G$ **do**
12:         $\hat{a}[i] \leftarrow \text{Linear}(h[i])$
13:     **end for**
14:     **return** $\{\hat{a}[i]\}_{i=1}^{N}$
15: **end procedure**

---

### 3.3.1 Initial features transformation

Before message propagation begins, each node in the graph is transformed via an encoder—based on an multilayer perceptron (MLP)—that projects the original features into a $d$-dimensional space, thereby facilitating the capture of non-linear relationships. The MLP employs a Tanh activation function to preserve both positive and negative values in the transformed features.

In other words, if $\vec{x_i} \in \mathbb{R}^{d_{in}}$ is the input feature vector for node $i$, then the encoder produces:

$$h_i^{\vec{(0)}} = f_{\text{node}}(\vec{x_i}), \quad f_{\text{node}} : \mathbb{R}^{d_{in}} \rightarrow \mathbb{R}^d \tag{2}$$

This process may incorporate regularization through dropout to stabilize training, although it is not employed in the present work.

Similarly, although it is not used in this paper, it is possible to enable a similar encoder via a configuration parameter to map edge attributes into a space compatible with dimension $d$. That is, if $\vec{e_{ij}}$ is the attribute associated with the edge between nodes $i$ and $j$, then:

$$\vec{z_{ij}} = f_{\text{edge}}(\vec{e_{ij}}), \quad f_{\text{edge}} : \mathbb{R}^{d_e} \rightarrow \mathbb{R}^d \tag{3}$$

As in the node encoder, a Tanh activation function can be applied in $f_{\text{edge}}$ to maintain the full range of values. This allows the additional information regarding the relationships between nodes to be integrated coherently with the node representations.

### 3.3.2 Message passing and aggregation

The architecture is built upon message passing layers based on *EdgeConv* Wang et al. (2019), a variant of graph convolutions that eases the integration of topological information from the nodes.

**Message function**   For each node $i$, a function $\phi$ is defined that combines the information from $i$ with that of each of its neighbors $j$. This function can be implemented using an MLP and takes into account both the current representations $h_i^{\vec{(l)}}$ and $h_j^{\vec{(l)}}$, as well as the edge attributes $\vec{z_{ij}}$—when available. The combination is expressed as:

$$m_{ij}^{\vec{(l)}} = \phi\left(h_j^{\vec{(l)}}, h_i^{\vec{(l)}}, \vec{z_{ij}}\right) \tag{4}$$

The function $\phi$ is designed to be invariant to the order of neighbors and to capture complex interactions. In our implementation, the MLP employs a Tanh activation function to maintain negative values

in the message representations, ensuring that the transformations preserve important directional information.

**Aggregation and update** The messages from all neighbours are aggregated using a summation function. The choice of this function is due to the properties it exhibits in the GNNs literature Gilmer et al. (2017); Xu et al. (2018).

$$\vec{m_i^{(l)}} = \sum_{j \in \mathcal{N}(i)} \vec{m_{ij}^{(l)}} \tag{5}$$

Where $\mathcal{N}(i)$ denotes the nodes in the neighborhood of node $i$. Next, the node is updated via a residual operation that concatenates the previous representation and the aggregated message, followed by a LayerNorm operation:

$$\vec{h_i^{(l+1)}} = \text{LayerNorm}\left(\text{concat}\left(\vec{h_i^{(l)}}, \vec{m_i^{(l)}}\right)\right) \tag{6}$$

This residual update—$\vec{h_i^{(l)}}$—helps preserve information across layers, while LayerNorm ensures stable training dynamics by normalizing the concatenated features.

**Regarding depth and receptive field** The number of layers $L$ is a critical hyperparameter: as $L$ increases, each node is able to access information from more distant neighbours in the graph, which is essential for capturing long-range interactions in $N$-bodies systems. However, an excessive number of layers may lead to over-smoothing Li et al. (2020), meaning that the node representations become overly homogeneous and the model's ability to differentiate between nodes is reduced, compounded by an increase in computational load that negates the original efficiency benefits.

### 3.3.3 Output layer and mapping to the regression task

The final module of the architecture is responsible for converting the representations obtained after the message passing layers into the desired output.

Each final node $i$ has a representation $\vec{h_i^{(L)}}$ which is transformed by a linear layer that maps the embedded space to a vector of dimension $d_{\text{out}}$. In our case, for the prediction of accelerations in a $3D$ space, $d_{\text{out}} = 3$:

$$\hat{\vec{a_i}} = W_{\text{out}} \vec{h_i^{(L)}} + \vec{b_{\text{out}}} \tag{7}$$

where $W_{\text{out}}$ and $\vec{b_{\text{out}}}$ are the parameters of the output layer.

### 3.4 Training and evaluation

After defining the dataset—through graphs derived from the raw simulation data—and the GNN architecture, the training and evaluation process aims to optimize both the model's predictive capability and generalization for the problem. Algorithm 4 describes the training process of the model.

The dataset has been divided with an approximately 90%-10% split, where 90% of the data is used for training and 10% for testing. No validation set was used. The loss function employed is the mean squared error (MSE) between the model's predicted accelerations and the actual accelerations.

For the training process, the Adam optimizer is used to update the model parameters. The implemented training cycle consists of:

- **Batch extraction**: Iterating over the training dataset in batches, where each batch contains a set of graphs along with their corresponding labels.
- **Inference**: For each batch, the model performs a forward pass through the graph, generating acceleration predictions $\vec{a_i}$ for each node.

**Algorithm 4** Training process

---

1: **procedure** TRAINGNN($D, E, B$)
2:     **for** epoch ← 1 **to** $E$ **do**
3:         Shuffle $D$
4:         Divide $D$ into $B$ batches
5:         **for** batch in $B$ **do**
6:             $loss \leftarrow 0$
7:             **for** graph $G$ in batch **do**
8:                 $\hat{a} \leftarrow$ ForwardPass($G$)
9:                 $loss \leftarrow loss + \text{MSE}(\hat{a}, Y)$
10:             **end for**
11:             Backpropagate $loss$
12:             Update weights (ADAM)
13:         **end for**
14:     **end for**
15:     Final test evaluation
16: **end procedure**

---

- **Loss calculation**: MSE is computed between the predictions and the actual accelerations.

- **Backpropagation**: The loss is backpropagated and the model weights are updated.

## 4 Experiments and results

The model was trained on a dataset comprising 60 random simulations, with 10 simulations for each of the following particle counts: 3, 25, 50, 100, 250, and 500. Each simulation ran for 1000 iterations. Evaluation was carried out on 6 additional simulations—one for each of the same particle counts—also executed over 1000 steps.

The simulations were carried out by modelling spiral galaxies that include a black hole. The simulation parameters are described in Table 1[1].

Table 1: Simulation parameters and their respective values

| Parameter | Value |
|---|---|
| Time step | 0.0001 |
| Gravitational constant | $4.5 \times 10^{-6}$ |
| Total galaxy mass | 1.0 |
| Radial scale | 3.0 |
| Vertical scale | 0.3 |
| Black hole mass proportion | 0.01 |
| Arms | 2 |

These parameters have been adjusted based on the total mass of the galaxy, in an effort to simulate the initial state of a galaxy in a dimensionless system in a relatively realistic manner. However, absolute fidelity was not the aim, as the focus of this paper is on the simulation of $N$-bodies systems; thus, while these values are representative, they do not correspond to any actual configuration of a theoretical or known galaxy.

### 4.1 Stepwise analysis

The results obtained from the model were grouped by scene. The data show that the loss is extremely low, with values on the order of $10^{-14}$ to $10^{-13}$. Figure 1 summarises the results for the loss and the average step times used during the simulation execution.

---

[1]The complete source code used in this work is publicly available at `https://github.com/KNODIS-Research-Group/efficient-n-body-simulations-using-physics-informed-gnn`.

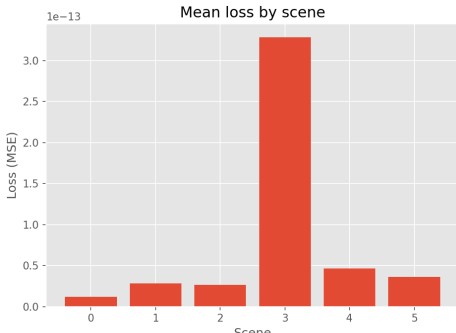
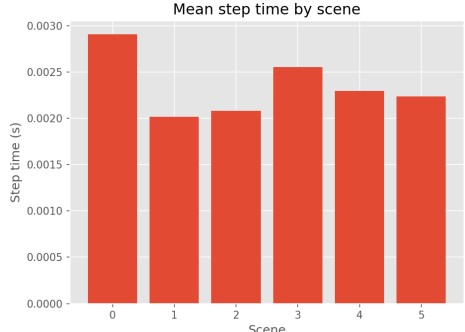

Figure 1: Results of the error analysis (left) and average scene execution time (right) for 1000 simulation steps performed using the model.

These results indicate that the model achieves more than acceptable accuracy and consistency in its predictions at each step.

## 4.2 Rollout evaluation

Errors for position, velocity, and acceleration of all particles have been averaged across scenes for each step. The acceleration error remains almost constant (with only minimal variation at the sixth decimal place), which indicates that the prediction of acceleration is very stable over time. In contrast, the errors in position and velocity start at zero and gradually increase, reaching values on the order of $10^{-21}$ and $10^{-17}$ respectively by the final step of the simulation, thereby demonstrating good control over error propagation in these variables. Figure 2 illustrates the evolution of these metrics over time.

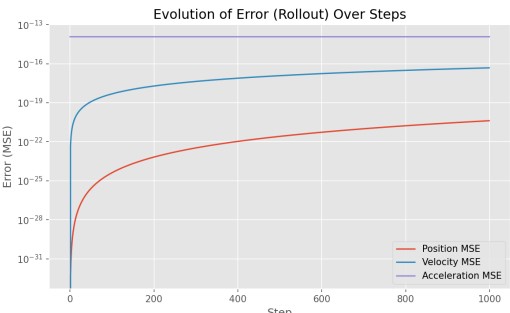

Figure 2: Evolution of the MSE in position, velocity, and acceleration over 1000 simulation steps. The value at each step is the average of the values for all particles across all scenarios for that step.

In the figure, one can observe the complete stability of the acceleration predictions and an almost imperceptible increase in the errors in position and velocity. Focusing on the accumulated errors (see Figure 3), which are obtained by summing the errors at each step, it reveals that:

1. The accumulated error in the position variable at the end of the simulation is on the order of $10^{-18}$, with a final average of approximately $8.27 \times 10^{-19}$.

2. The accumulated error in velocity is on the order of $10^{-14}$, with a final average of approximately $1.6 \times 10^{-14}$.

3. The accumulated error in acceleration reaches values on the order of $10^{-11}$, with a final average of approximately $1.26 \times 10^{-11}$.

These low error values indicate that the error propagation over time is practically negligible.

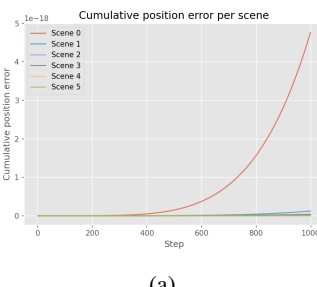 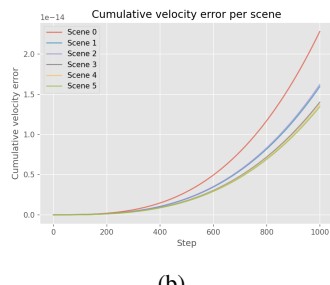 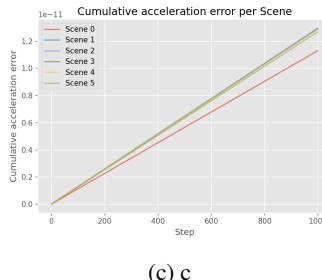

| (a) | (b) | (c) c |

Figure 3: Accumulated errors in position (left), velocity (centre) and acceleration (right). The error accumulation is shown for each of the 6 scenes, from the one with the fewest particles (scene 0, 3 bodies) to that with the most particles (scene 5, 500).

### 4.3 Regarding computational efficiency

A speedup of approximately 1.17 times has been achieved, that is, the model runs roughly 17% faster than the base simulation. Although moderate, it represents a promising avenue for further research, particularly as it has been realised without sacrificing the accuracy of the predictions.

## 5 Conclusion

Following the analysis of the results obtained in this study, it appears that integrating GNNs techniques with physical $N$-bodies simulations can accelerate their execution while maintaining a high degree of accuracy, with very low losses and negligible accumulated errors.

It should be noted that the dataset was divided into a training-test split without a dedicated validation set. This decision was primarily driven by the focus on establishing baseline performance. However, the absence of a validation set is a limitation, as it restricts our ability to monitor overfitting. Future work will aim to incorporate a separate validation set or adopt cross-validation techniques to enhance performance evaluation and robustness.

Another relevant consideration is that the black hole is treated uniformly as any other particle, which may inadequately capture its dominant gravitational influence. A differentiated treatment—such as assigning a higher weight or designing a specialized connectivity scheme—should be explored in future work to better reflect its unique role in the system. Additionally, further improvements to the implementation may enhance the model's performance and make it even more competitive compared to traditional simulation techniques.

Finally, although this study focuses on small- to medium-scale systems (up to 500 bodies), real-world astrophysical simulations often involve thousands or even millions of particles. Addressing scalability is therefore a key avenue for future research. Possible approaches include using sparse graph representations, hierarchical modeling strategies, or hardware acceleration to extend the applicability of the method to larger and more complex systems.

### Acknowledgments and Disclosure of Funding

The support of the Comunidad Autónoma de Madrid under ALENTAR-J-CM project (reference TEC-2024/COM-224) is gratefully acknowledged.

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
