# OpenReview forum: "Efficient n-body simulations using physics informed graph neural networks"
_MAEB/2025/Congreso — MAEB 2025_

### Official Review · Reviewer_H8bT · 2025-03-17
**Combining DL with simulation-based techniques is an evolving field and relevant to the current SoA**

**Rating:** 5
**Confidence:** 5

**Review:**

In this research work the GNN model's ability to accurately predict accelerations is evaluated in N-body systems, demonstrating that integrating realistic physical simulations with deep learning can significantly enhance simulation speed while preserving a high level of fidelity to real-world dynamics.

This paper demonstrates strong scientific quality, presenting clear, well-structured findings and are relevant to the field.

Pros:

The experimental results show that it is a promising research line as the proposed approach runs 17% faster than the simulation.
The methodology is clearly explained, providing all necessary details for implementation.

Cons:

Line 145 the algorithm enumeration is missing
The results may not be robust as they did not use a validation set

---

### Official Review · Reviewer_uTTQ · 2025-03-18
**GNNs as an efficient computational shortcut**

**Rating:** 4
**Confidence:** 3

**Review:**

The paper combines graph neural networks (GNNs) with classical n-body simulation methods to accelerate the computation of gravitational interactions. A leapfrog integration scheme generates simulation data from diverse astrophysical scenarios. This data is then converted into graph representations where each particle is a node and edges are determined via a k-nearest neighbors criterion. The results demonstrate that the model achieves low prediction errors in scenarios ranging from 3 to 500 bodies, and a moderate speedup.

In essence, GNNs serve as an efficient computational shortcut within a traditional simulation framework. Thus, while promising, they are not “new” numerical methods in the classical sense (like finite element or finite difference methods), but rather a modern tool from deep learning that enhances simulation efficiency by learning approximations.

The experiments are conducted on a dataset comprising 60 training simulations and 6 testing simulations, spanning n-body scenarios from 3 to 500 bodies over 1000 time steps. This range is sufficient to demonstrate the feasibility and accuracy of the approach on small to moderately sized systems. In any case, there are scalability concerns because real-world astrophysical simulations often involve many more particles (potentially thousands to millions) to capture the full complexity of gravitational systems. The experimental scale in the paper, while adequate for initial validation, is relatively modest compared to such large-scale applications.

---

### Official Review · Reviewer_7sAE · 2025-03-19
**Using GNNs to accelerate complex n-bodies simulations with high precision**

**Rating:** 5
**Confidence:** 3

**Review:**

The paper presents an approach to predict particle acceleration in N-bodies simulations making use of physics-informed GNNs and traditional numerical methods. The paper is well written, easy to understand and follow. The problem addressed is very interesting and have important applications.Also, the method proposed to solve the problem, that combines deep learning with traditional numerical methods is very interesting and shows the possibilities that DL offer to speedup computations in this kind of problems.

The method is correctly evaluated in random scenarios with different features, and results show very low prediction errors and a notable speedup thanks to the use of the proposed GNN.

Please, proofread the paper. For instance, there is a broken reference in line 145.

---

### Decision · Program_Chairs · 2025-03-20

Accept